Identification of key genes of human bone marrow stromal cells adipogenesis at an early stage

Chen Pengyu 1 2
Song Mingrui 1 2
Wang Yutian 1 2
Deng Songyun 1 2
Hong Weisheng 1 2
http://orcid.org/0000-0002-0992-5013 Zhang Xianrong 1 2 xianrongzh@smu.edu.cn
Yu Bin 1 2 yubin@smu.edu.cn
1 Department of Orthopaedics, Nanfang Hospital, Southern Medical University , Guangzhou, Guangdong , China
2 Guangdong Provincial Key Laboratory of Bone and Cartilage Regenerative Medicine, Nanfang Hospital, Southern Medical University , Guangzhou, Guangdong , China
Nakai Kenta
Electronic publication date: 2020 Jul 21
Publication date: 2020
Volume: 8
Electronic Location ID: e9484
Received 2020 Mar 20; Accepted 2020 Jun 15
Copyright: © 2020 Chen et al.
Copyright year: 2020
Copyright holder: Chen et al.
License: This is an open access article distributed under the terms of the Creative Commons Attribution License, which permits unrestricted use, distribution, reproduction and adaptation in any medium and for any purpose provided that it is properly attributed. For attribution, the original author(s), title, publication source (PeerJ) and either DOI or URL of the article must be cited.
License URL: https://creativecommons.org/licenses/by/4.0/

Keywords: Adipogenesis, Bone marrow stromal cells, Bone marrow adipocytes, Bioinformatics, Differentially expressed genes

Funding: National Natural Science Foundation of China 81772366 and 81830079 This study was supported by grants from the National Natural Science Foundation of China (No. 81772366) and The Major Program of National Natural Science Foundation of China (No. 81830079). The funders had no role in study design, data collection and analysis, decision to publish, or preparation of the manuscript.

==============================
Background

Bone marrow adipocyte (BMA), closely associated with bone degeneration, shares common progenitors with osteoblastic lineage. However, the intrinsic mechanism of cells fate commitment between BMA and osteogenic lineage remains unclear.

Methods

Gene Expression Omnibus (GEO) dataset GSE107789 publicly available was downloaded and analyzed. Differentially expressed genes (DEGs) were analyzed using GEO2R. Functional and pathway enrichment analyses of Gene Ontology and Kyoto Encyclopedia of Genes and Genomes were conducted by The Database for Annotation, Visualization and Integrated Discovery and Gene set enrichment analysis software. Protein–protein interactions (PPI) network was obtained using STRING database, visualized and clustered by Cytoscape software. Transcriptional levels of key genes were verified by real-time quantitative PCR in vitro in Bone marrow stromal cells (BMSCs) undergoing adipogenic differentiation at day 7 and in vivo in ovariectomized mice model.

Results

A total of 2,869 DEGs, including 1,357 up-regulated and 1,512 down-regulated ones, were screened out from transcriptional profile of human BMSCs undergoing adipogenic induction at day 7 vs. day 0. Functional and pathway enrichment analysis, combined with modules analysis of PPI network, highlighted ACSL1, sphingosine 1-phosphate receptors 3 (S1PR3), ZBTB16 and glypican 3 as key genes up-regulated at the early stage of BMSCs adipogenic differentiation. Furthermore, up-regulated mRNA expression levels of ACSL1, S1PR3 and ZBTB16 were confirmed both in vitro and in vivo.

Conclusion

ACSL1, S1PR3 and ZBTB16 may play crucial roles in early regulation of BMSCs adipogenic differentiation

Introduction

Bone marrow adipocytes (BMA) were discovered beyond a century ago, but our knowledge about their origin, functions and roles in local bone marrow is still limited (Horowitz et al., 2017). BMA is now gradually acknowledged as a distinct group of adipose tissue different from white, brown and beige adipose tissues (Chen et al., 2014). Studies have shown that BMA is different from extramedullary adipose tissue in such aspects as origin, location, adipocyte size, content of fatty acids, cytokine and adipokine expression, stem cell markers expression and immunomodulatory properties. Recently, BMA is considered to interact with and influence other cell populations in and out of bone marrow, playing a crucial role in local bone homeostasis and whole body energy metabolism (Nuttall et al., 2014) rather than simply filling cavities of bones. BMA takes part in various pathophysiological conditions such as aging, obesity, osteoporosis, estrogen deficiency (Devlin & Rosen, 2015; Justesen et al., 2001). Moreover, BMA is negatively correlated with bone regeneration (Ambrosi et al., 2017) of stem cells in local marrow environment, likely through paracrine and endocrine of specific classes of cytokine and adipokine (Yokota et al., 2000).

Bone marrow stromal cells (BMSCs) are known as a source of BMA progenitors. Studies have identified that several BMSCs subpopulations such as osterix+ cells, leptin receptor (LepR)+ cells, Sca1+ cells and nestin+ cells were capable of differentiating into adipocytes (Ambrosi et al., 2017; Ono et al., 2014; Zhou et al., 2014). Meanwhile, they are multipotent cells that are able to give rise to osteoblasts, chondrocytes, fibroblasts and marrow stromal cells (Kassem & Bianco, 2015), indicating that these aforementioned cells, especially osteoblasts, may share common progenitors with adipocytes (Mizoguchi et al., 2014). Studies linking adipogenesis to osteogenesis consolidated this possibility. Conditional LepR deletion in BMSCs using Prx1-Cre recombinase led to inhibited adipogenesis and accelerated fracture healing (Yue et al., 2016). Conversely, PTH1R deletion in mesenchymal stem cells (MSCs) led to formation of adipocytes (Fan et al., 2017), and mice treated with Bisphenol-A-diglycidyl ether, a PPARγ antagonist, showed higher levels of osteoblastogenesis and bone formation concomitant with decreased marrow adiposity and ex vivo adipogenesis (Duque et al., 2013). However, no existing evidence can directly prove a causal relationship between BMA and bone loss (Li, Wu & Kang, 2018). Therefore, further investigations of intrinsic mechanism underlying adipogenesis of BMSCs at an early stage may provide new insight into BMSCs lineage commitment.

Recently, high-throughput techniques such as microarray assay have been widely used to analyze expression of genes associated with pathophysiological processes of diseases, thereby identifying target genes which may be further applied in diagnosis and prognosis prediction (Rung & Brazma, 2013). In the present study, dataset GSE107789 from Gene Expression Omnibus (GEO) was analyzed with differentially expressed genes (DEGs), Gene Ontology (GO) terms, Kyoto Encyclopedia of Genes and Genomes (KEGG) pathways, Gene set enrichment analysis (GSEA) and Protein–Protein interaction (PPI) analysis.

Materials and Methods

Expression profile data

The raw gene profile dataset GSE107789, based on the platform of Agilent-028004 SurePrint G3 Human GE 8 × 60 K Microarray chip (GPL14550), was downloaded from GEO database (http://www.ncbi.nlm.nih.gov/geo/). The data were from human BMSCs line (hMSC-TERT) undergoing adipogenic differentiation on day 7. Total RNA of three replicates from control and adipocytic differentiation group was extracted and hybridized to the one-color Agilent Human GE 8 × 60 K Microarray chip.

Analysis of DEGs

Based on the data from the GEO dataset GSE107789, we analyzed DEGs in BMSCs undergoing adipogenic differentiation vs. BMSCs treated with control agent on day 7. Significant values for DEGs were obtained using web analysis tool GEO2R (https://www.ncbi.nlm.nih.gov/geo/geo2r/) (Barrett et al., 2013). The adjusted p values were calculated using Benjamini and Hochberg false discovery rate method by default. Hence, fold changes were transformed to log2 values for volcano plot. |Logarithm of fold change (LogFC)| cutoff > 1.25 and adjusted p < 0.01 were used to determine significantly changed transcripts.

Functional and pathway enrichment analysis

Gene Ontology terms, including biological process (BP), cellular component (CC), and molecular function (MF), have been widely used for gene annotation and characterization of high-throughput genome or transcriptome data (Ashburner et al., 2000). KEGG is a website database for pathways enrichment analysis of target genes sets, to obtain integration and interpretation of these data sets (Kanehisa et al., 2012). The Database for Annotation, Visualization and Integrated Discovery (DAVID, https://david.ncifcrf.gov/) was used to identify GO terms and KEGG pathway enrichment analysis of DEGs (Da Huang, Sherman & Lempicki, 2009). Moreover, GSEA software (GSEA 4.0.3, https://www.gsea-msigdb.org/gsea/index.jsp) was used to analyze the dataset as well. KEGG subset of canonical pathway (c2.cp.kegg.v7.0) from Molecular Signatures Database was used for analysis and gene sets with normalized enrichment score >1 and normalized p < 0.01 were considered as statistical significance. Leading edge analysis was carried out to screen out the key genes.

PPI network analysis

Protein–protein interactions network analysis was performed in order to identify crucial proteins and protein modules involved in BMSC adipogenic differentiation. In the present study, STRING database (http://www.string-db.org/) was used to converse DEGs to corresponding protein and PPI network (Szklarczyk et al., 2015). In consideration of the 2,000 genes count limit of STRING, DEGs set containing 1,668 genes with |LogFC| > 2 was put into analyzing. Cytoscape (https://cytoscape.org/), an open source software platform for visualizing molecular interaction networks and biological pathways and integrating these networks with annotations, gene expression profiles and other state data, was used to visualize and analyze topologic parameters, including degree of distribution, closeness centrality and betweenness centrality of PPI data obtained from STRING database. Nodes (standing for corresponding proteins) with a higher degree which form more edges (standing for PPIs) with other proteins were defined as hub proteins. Moreover, module analysis of the present PPI network was performed, using MCODE, a plugin of Cytoscape, with the following settings: degree cutoff = 2, node score cutoff = 0.2, k-core = 2, and max. depth = 100 (Bader & Hogue, 2003). Interested modules were screened out, manually by MCODE grade and proteins involved in the corresponding module, and further investigated by GO terms and KEGG pathway enrichment analysis.

Experimental animals

C57BL/6J mice were provided by the Experimental Animal Center at Southern Medical University, Guangzhou, China. All animal protocols were approved by the Animal Care and Use Committee at Nanfang Hospital, Southern Medical University (NFYY-2018-34) and all animal experiments were performed in accordance with the guidelines of Animal Care and Use Committee. Mice were raised under specific pathogen-free conditions at 18–28 °C with humidity of 40–70%, free access to food and water with a 12 h light/dark cycle. 12-week-old mice were randomly divided into two groups with bilateral ovariectomy (OVX) and sham surgery. At week 8 after the surgery, mice were euthanized and bone samples were harvested for further analysis. Bone marrow of tibias and femurs from the left side was flushed out and stored at −80 °C for mRNA expression analysis. Tibias and femurs from the right side were dissected free of soft tissue and processed for histological analysis.

Isolation and culture of BMSCs

Bone marrow stromal cells were isolated from 4–6 week-old mice following the protocol previously described (Maridas et al., 2018). Briefly, after both ends of tibia and femur were removed, bone marrow was flushed out with α-MEM under sterile condition. Flushed cells were passed through 70 μm cell strainer and seeded at a density of 1 × 106 cells/cm2 in 100 mm dish for expanding. Cells were cultured with growth medium (α-MEM with 15% fetal bovine serum, 55 μM β-mercaptoethanol, 2 mM glutamine, 100 IU/ml penicillin and 100 μg/ml streptomycin). For adipogenic differentiation, cells were seeded at a density of 2 ×105 cells/ cm2 in six-well plates with growth medium. When cells reach 100% confluence, medium was replaced with adipogenic differentiation media (MUBMX-03031, Cyagen Biosciences Inc., China) following manufacturer’s instructions. Cells were harvested for mRNA expression analysis or Oil red staining at day 7 after adipogenic differentiation.

Oil red O staining

Lipid droplets were detected by Oil Red O (ORO) staining. BMSCs were washed with PBS and fixed with 4% paraformaldehyde for 30 min at room temperature, washed with distilled water and 60% isopropanol, and then incubated with ORO staining solution (0.3% ORO powder in 60% isopropanol, Sinopharm Chemical Reagent Co. Ltd., China) for 30 min at room temperature. For ORO staining in bone tissue, tibias were fixed, decalcified followed by frozen embedding. Sections at 8-μm were used for ORO staining. Images were acquired under a BX53 microscope (OLYMPUS Co., Japan). Quantification of ORO was obtained by ImageJ software. For cell culture experiments, five different fields were randomly imaged and counted for ORO-positive cells and the total cells. For frozen sections, the ratio of ORO-positive staining area to the total area of each photograph was calculated.

RNA isolation and real-time quantitative PCR analysis

Total RNA was isolated using RNAiso Plus (Takara, Kyoto, Japan) and reverse-transcribed using the PrimeScript RT reagent Kit (Takara, Kyoto, Japan) according to the manufacturer’s instructions. The cDNA samples were then used for real-time quantitative PCR (qRT-PCR) analysis with TB Green Premix Ex Taq II (Takara, Kyoto, Japan) following the manufacturer’s instructions. Relative expression was calculated using 2−ΔΔCt method and normalized with GAPDH. The primer sequences of corresponding genes used in qRT-PCR analysis are listed in Table 1.

Table 1 Primer sequences for qRT-PCR.

Gene symbol	Forward primers	Reverse primers	
GAPDH	TGTCGTGGAGTCTACTGGTG	GCATTGCTGACAATCTTGAG	
S1PR3	TTATGTCCGGCAGGAAGACG	ATCATGGTCAGGTGTCGCTC	
ACSL1	TGCCAGAGCTGATTGACATTC	GGCATACCAGAAGGTGGTGAG	
ZBTB16	CTGGGACTTTGTGCGATGTG	CGGTGGAAGAGGATCTCAAACA	
GPC3	CAGCCCGGACTCAAATGGG	GCCGTGCTGTTAGTTGGTATTTT	
ADIPOQ	GTTCCCAATGTACCCATTCGC	TGTTGCAGTAGAACTTGCCAG	
CEBPA	TTCGGGTCGCTGGATCTCTA	TCAAGGAGAAACCACCACGG	
PPARG	TCGCTGATGCACTGCCTATG	GAGAGGTCCACAGAGCTGATT	

Statistical analysis

All statistical analyses were performed using GraphPad Prism 7.0 (GraphPad software, Inc., La Jolla, CA, USA) with Student’s t-test. All experiments in BMSCs were repeated independently at least three times. All data were represented as mean ± SEM. Values of p < 0.05 were considered statistically significant.

Results

Identification of DEGs

We screened out 2,869 DEGs using GEO2R, including 1,357 up-regulated genes and 1,512 down-regulated genes, based on the dataset GSE107789. Corresponding volcano plot is shown in Fig. 1 and heat map of top 100 DEGs in Fig. 2.

Figure 1 Volcano plot of −log10 (adjusted p-value) vs. log2 (fold change).

Red plots represent significant DEGs determined by |log2 (fold change)| > 1.25 and adjusted p value < 0.01.

Figure 2 Heat map of top 100 DEGs.

The bar indicate expression level of DEGs. Ad, adipogenesis; Con, control.

GO terms analysis

To get a better insight into how DEGs are orchestrated in the process of BMSC adipogenic differentiation, DAVID database was used to perform GO terms enrichment analysis. For Biological Functions (BP), we identified 38 GO terms. The enriched terms were mostly related to cell cycle and mitosis, such as cell division, DNA replication, mitotic nuclear division, G1/S transition of mitotic cell cycle and DNA replication-dependent nucleosome assembly (Fig. 3A). For Cell Content (CC), DEGs were enriched in 21 terms while the top terms were mostly related to chromosome and extracellular content, such as chromosome space, nuclear chromosome and extracellular region. For MF, we identified seven GO terms, including protein binding, protein heterodimerization activity, heparin binding, single-stranded DNA-dependent ATPase activity, growth factor activity, cytoskeletal protein binding, histone binding.

Figure 3 GO terms analysis of DEGs.

Top 10 GO terms of all DEGs (A) and top 200 DEGs (B) enriched in biological process, cellular component and molecular function were shown, respectively. Color guide indicates −log10 (p value) of GO terms. Gene counts represent number of genes enriched in the corresponding terms.

Gene Ontology terms were further analyze for the top 200 DEGs. As shown in Fig. 3B, the top 10 BP terms enriched were skeletal system development, regulation of cell proliferation, response to lipopolysaccharide, aging, female pregnancy, brown fat cell differentiation, inflammatory response, positive regulation of cell migration, cell adhesion and extracellular matrix organization.

KEGG pathway analysis

DAVID database was used to obtain KEGG pathway enrichment. DEGs count and enriched KEGG pathway of all DEGs are shown in Fig. 4A. The top five pathways were DNA replication, cell cycle and systemic lupus erythematosus, mismatch repair and Alcoholism. In addition, KEGG pathway analysis of top 200 DEGs was performed (Fig. 4B). The top five pathways enriched were pathways in cancer, tyrosine metabolism, drug metabolism-cytochrome P450, PPAR signaling pathway and PI3K-Akt signaling pathway.

Figure 4 KEGG pathway analysis of DEGs.

Color of bars represent −log10 (p value) while x axis represent number of genes enriched in the corresponding pathway. p value < 0.05 was considered significant enrichment for all DEGs (A) and top 200 DEGs (B).

GSEA enrichment analysis

A total of 19 significant enriched pathways were identified by GSEA (as shown in Table 2), three of which were closely associated with adipogenesis or lipogenesis (Figs. 5A–5C). Furthermore, leading edge analysis was conducted of these three pathways (Figs. 5D and 5E) in which 3 of 51 genes, CPT1A, ACSL1 and ACSL3, appeared in all three pathways (Fig. 5D). Most importantly, ACSL1 had the highest transcriptional level in adipogenic cells compared to CPT1A and ACSL3 (Fig. 5E).

Figure 5 GSEA analysis of GSE107789 highlighted ACSL1 as candidate gene.

Pathways associate with adipogenesis or lipogenesis enriched by GSEA analysis, which were PPAR signaling pathway (A), fatty acid metabolism (B) and adipocytokine signaling pathway (C), respectively. Leading edge analysis of above mentioned pathways (D and E). Genes appear in aforementioned pathways (D). X-axis represents number of gene sets the corresponding gene appear in. The bolder line represents ACSL1. Corresponding heat map were shown and ACSL1 was emphasize by bolder font (E). The color guide indicated the rank score of genes.

Table 2 Pathways enriched in adipogenic induced BMSCs by GSEA analysis.

NAME	NES	NOM p-value	FDR q-value	
Ribosome	2.77	0	0	
Drug metabolism cytochrome p450	2.01	0	0.003	
Metabolism of xenobiotics by cytochrome p450	2	0	0.002	
Nitrogen metabolism	1.85	0	0.012	
Arachidonic acid metabolism	1.8	0	0.02	
Glycolysis gluconeogenesis	1.78	0	0.022	
Steroid hormone biosynthesis	1.73	0	0.025	
PPAR signaling pathway	1.69	0	0.032	
Fatty acid metabolism	1.66	0	0.036	
Linoleic acid metabolism	1.65	0	0.035	
Olfactory transduction	1.38	0	0.146	
Sulfur metabolism	1.77	0.002	0.021	
Renin angiotensin system	1.9	0.002	0.007	
ECM receptor interaction	1.62	0.003	0.04	
Lysosome	1.61	0.003	0.039	
Cytokine cytokine receptor interaction	1.44	0.003	0.115	
Complement and coagulation cascades	1.66	0.005	0.035	
Adipocytokine signaling pathway	1.61	0.005	0.039	
Glycosaminoglycan biosynthesis chondroitin sulfate	1.7	0.007	0.031	
Glutathione metabolism	1.62	0.009	0.041	
Note:

NES, normalized enrichment score; NOM, nominal; FDR, false discovery rate. Gene sets with NOM p value < 0.01 are considered as statistical significance.

PPI network analysis: hub proteins and PPI module

Protein–protein interactions network was obtained using STRING database. In consideration of a huge number of DEGs, we excluded out of the PPI network analysis the DEGs whose |logFC| was less than 2, leaving 1,668 DEGs for further analysis. As shown in Fig. 6A, 1,481 nodes and 21,274 edges were included in the network. Among proteins constructing the present network, those with high node degrees were considered as hub proteins. In this case, Cyclin-dependent kinase 1 (CDK1), Cyclin-A2 (CCNA2), G2/mitotic-specific cyclin-B1, aurora kinase A (AURKA), aurora kinase B (AURKB), CDC20, PLK1, BUB1, TOP2A, interleukin 6 (IL6) were the top 10 hub proteins. In pursuit of a better understanding of the results of PPI network analysis, MCODE plugin of Cytoscape was introduced to screen out top PPI modules. The top three modules are shown in Figs. 6B–6D, respectively. Module 1 of the present PPI network was consisted of 142 nodes and 8,251 edges. Most of the proteins clustered in module 1 were down-regulated during adipogenic differentiation. The top five proteins in top three modules are shown in Table 3. The top five proteins with higher degrees in module 1 were CDK1, CCNA2, CCNB1, AURKA and AURKB. Module 2 was consisted of 59 nodes and 678 edges, including interleukin 8 (IL8), Stromal cell-derived factor 1 and sphingosine 1-phosphate receptors 3 (S1PR3). Moreover, module 3, which comprised 50 nodes and 358 edges, contained IL6, bone morphogenetic protein 4 (BMP4), apolipoprotein B-100 (APOB), osteopontin (SPP1), zinc finger and BTB domain containing 16 (ZBTB16), glypican 3 (GPC3), etc.

Figure 6 PPI analysis of DEGs.

(A) Whole PPI network of DEGs. (B) Module 1. (C) Module 2. (D) Module 3. The nodes represent DEGs, whereas lines between the nodes represent interactions between DEGs. Moreover, sizes of nodes represent degrees of nodes while the color represent closeness centrality as shown by the colored bar above.

Table 3 Top five hub proteins in each top three module.

Cluster	Name	Degree	Betweenness centrality	Closeness centrality	Log2 (Fold change)	
Module 1	CDK1	253	0.022	0.428	−2.344	
CCNA2	227	0.011	0.421	−2.726	
CCNB1	226	0.016	0.423	−3.126	
AURKA	216	0.014	0.412	−3.33	
AURKB	215	0.007	0.403	−3.583	
Module 2	CXCL8	115	0.017	0.409	5.541	
CXCL12	95	0.009	0.385	2.184	
DNMT1	87	0.009	0.393	−2.166	
AGT	83	0.011	0.375	7.155	
HIST1H2BK	80	0.003	0.36	2.13	
Module 3	IL6	203	0.086	0.445	−2.289	
BMP4	88	0.015	0.393	−3.008	
APOB	79	0.014	0.378	4.305	
SPP1	75	0.007	0.397	5.079	
HIST1H4A	73	0.004	0.369	−2.533	

Verification of candidate gene expression

Based on the above data, four candidate genes, S1PR3, ACSL1, ZBTB16 and GPC3, were further verified in adipogenic cells and OVX osteoporosis mice model. For adipogenic differentiation of BMSCs, ORO staining showed more lipid drops in cells undergoing adipogenic differentiation on day 7 (Figs. 7A and 7B). qRT-PCR analysis showed up-regulated expression of adipogenic marker genes (ADIPOQ, CEBPA, PPARG) (Fig. 7C), confirming the adipogenic differentiation of BMSCs. Next, we verified the expression of the candidate genes. Results showed significantly increased mRNA expression of ACSL1, S1PR3 and ZBTB16 (Fig. 7D).

Figure 7 Verification of the mRNA expression of candidate genes.

(A) Representative images of Oil Red O (ORO) staining for adipogenic differentiation of BMSCs. Scale bar: 100 μm. (B) Quantification of the ratio of ORO-positive stained cells to total cell number. N = 3/group. (C) qRT-PCR analysis of mRNA expression of adipogenic markers (CEBPA, PPARG and ADIPOQ). (D) qRT-PCR analysis of mRNA expression of candidate genes (ACSL1, S1PR3, ZBTB16 and GPC3). N = 3/group. BMSCs were cultured in growth medium (control) or adipogenic differentiation medium for 7 days. Total RNA were collected for genes expression analysis. (E) Representative images of ORO staining for frozen sections of proximal tibia from control and OVX mice. Scale bar: 200 μm. (F) Quantitative analysis of the ratio of ORO staining area to tissue area. N = 4/group. (G) qRT-PCR analysis of mRNA expression levels of candidate genes in bone marrow of sham and OVX mice (n = 5). Data are represented as means ± SEM. Adi, BMSCs cultured in adipogenic differentiation.*p < 0.05, **p < 0.01.

Further, OVX mice model was used to verify the alterations of the above four candidate genes in vivo. ORO staining showed more lipid drops in tibias from OVX mice (Figs. 7E and 7F), confirming the increased adipogenesis. qRT-PCR confirmed up-regulated expression of ACSL1, S1PR3 and ZBTB16 in bone (Fig. 7G). Together, the above data suggest intrinsic connections of ACSL1, S1PR3 and ZBTB16 with adipogenesis.

Discussion

Accumulation of BMA is accompanied with diseases and such pathological conditions as obesity, aging and osteoporosis (Devlin & Rosen, 2015). It has been demonstrated that BMA shares common progenitors with osteogenic lineage (Ambrosi et al., 2017; Ono et al., 2014; Zhou et al., 2014). However, the intrinsic mechanism of cells fate commitment between BMA and osteogenic lineage is still a mystery. Global gene expression profiling by Ali et al. (2018) using microarray assay on day 7 after adipogenic differentiation of hMSC-TERT identified FAK and IGF1R signaling as important pathways during bone marrow adipogenesis. However, intrinsic connection of DEGs remains unexplored. Here we identified 2,869 DEGs using publicly available data GSE107789. Additionally, GO terms and KEGG pathway enrichment, GSEA analysis and PPI network analysis identified four genes (ACSL1, S1PR3, ZBTB16 and GPC3) which may be closely associated with early stage adipogenesis of BMSCs. Further we demonstrated up-regulated expression of ACSL1, S1PR3 and ZBTB16 in vitro and in vivo.

Gene Ontology terms analysis revealed a variety of functional categories. Cell division, DNA replication, mitotic nuclear division, G1/S transition of mitotic cell cycle and DNA replication-dependent nucleosome assembly were the top five biological function (BP) terms related to cell cycle and cell division. The present study found that genes involved in these categories were mostly down-regulated, in agreement with previous studies (Ali et al., 2018; Menssen et al., 2011). In order to obtain more significant GO terms, we performed analysis on the top 200 DEGs with lower p values. As a result, skeletal system development, regulation of cell proliferation, response to lipopolysaccharide, aging, female pregnancy, brown fat cell differentiation, inflammatory response, positive regulation of cell migration, cell adhesion, extracellular matrix organization were the top 10 BP terms, indicating that these processes are highly associated with adipogenesis. This is consistent with our hypothesis that BMA might be closely connected with bone homeostasis. Simultaneously, KEGG pathway analysis highlighted such pathways correlated with adipogenesis as PI3K-Akt signaling pathway, Focal adhesion pathway, FoxO signaling pathway, p53 signaling pathway and PPAR signaling pathway, similar to previous findings (Ali et al., 2018).

Gene set enrichment analysis analysis highlighted ACSL1 as a potential key gene in regulating adipogenic differentiation. ACSL1, which belongs to the class of acyl-CoA synthetases, plays a crucial role in fatty acid metabolism and lipid synthesis as well (Lobo, Wiczer & Bernlohr, 2009). It is highly expressed in adipogenic cells, hypothesized to be regulated by PPARs (Martin et al., 1997). Consistently, our work demonstrated up-regulated expression of ACSL1 during adipogenesis and in osteoporotic bone. However, it remains unclear and deserves further investigation whether ACSL1 might regulate BMSCs commitment or differentiation.

The PPI network was highly clustered in the present study. Module 1, consisting of CDK1, CCNA2, CCNB1, AURKA, AURKB and so on, is associated with cell cycle and mitosis. These proteins may also play a role in adipogenesis and MSCs commitment (Marquez et al., 2017; Park et al., 2011). Module 2 is interesting as it contains several chemokines and chemokines receptors include IL8, CXCL12 and S1PR3 which is one of the 5 receptors of sphingosine 1-phosphate (S1P). S1P receptors (S1PRs) have been demonstrated to participate in differentiation processes of different cell types including myoblast, epicedial progenitor and MSCs (Bruno et al., 2015; Li et al., 2019; Price et al., 2015). Additionally, evidence has shown that S1PR3 mediates fibroblast differentiation, as well as BMSC migration and proliferation (Li et al., 2009; Price et al., 2015). Based on our results of PPI module analysis, we hypothesized that S1PR3 might regulate BMSCs differentiation in a similar manner with chemokines such as IL8 and CXCL8 or in collaboration with chemokine signaling pathway. However, further study is required to demonstrate the exact role of S1PR3 in BMSCs differentiation.

Gene Ontology analysis of module 3 revealed such categories associated with the skeletal system as embryonic hindlimb morphogenesis, skeletal system development and positive regulation of ossification, enriched by genes like ZBTB16 and GPC3 and along with BMP4 (Shown in Data S1), suggesting that these genes functioned in adipogenesis and osteogenesis as well, probably through cooperation with BMP4.

Zinc finger and BTB domain containing 16 gene, a member of the Krueppel C2H2-type zinc-finger protein family, encodes a BTB/POZ domain and zinc finger containing transcription factor known as PLZF (Li et al., 1997). In line with the present study, ZBTB16 was identified as a potential early and late-stage differentiation marker during adipogenesis in human adipose-derived stromal cells (Ambele et al., 2016). Wei et al. (2018) reported that ZBTB16 overexpression promoted white adipogenesis and induced brown-like adipocyte formation for bovine white intramuscular preadipocytes. Besides, it was reported that zoledronic acid accelerated the MSCs differentiation to the osteoblast cells through promotion of the ZBTB16 expression (Marofi et al., 2019). Agrawal Singh et al. (2019) proposed that ZBTB16 gene expression is induced at an early stage of hMSCs differentiation and might promote naive stem cells commitment, possibly through positive regulation of enhancer function. Thus, up-regulation of ZBTB16 in an early stage of both adipogenic and osteogenic differentiation of hMSC may imply a potential role of ZBTB16 as a promoter or enhancer of hMSC commitment in multiple lineages. However, the explicit role of ZBTB16 in hMSCs adipogenic differentiation and lineage commitment remains undetermined.

Ovariectomy disturbed BMSCs differentiation, leading to a more adipogenic status (Liu et al., 2016). Consistently, ORO staining in the present study showed a higher BMA content in OVX mice, indicating that adipogenesis in bone marrow was increased. Based on these findings, we used OVX mice as an in vivo model for verification of candidate genes. qRT-PCR confirmed expression profiles of ACSL1, S1PR3 and ZBTB16 both in vitro and in vivo, implying potential functions of these genes in early adipogenesis or at the onset of BMSCs differentiation.

Conclusion

In conclusion, 2,869 DEGs have been identified, including 1,357 up-regulated genes and 1,512 down-regulated genes. Further investigation of DEGs found that the genes regulating osteogenesis were involved in adipogenesis, implying potential roles of these genes in hMSCs lineage commitment and adipogenesis initiation. GSEA and PPI analysis highlighted ACSL1, S1PR3, ZBTB16 and GPC3 among these genes. Furthermore, ACSL1, S1PR3 and ZBTB16 were verified by qRT-PCR at transcriptional level. Overall, the present study suggests a close correlation between these genes and early stage adipogenesis or onset of BMSCs differentiation. Further investigation and prudent verification are required for a better understanding of their functions.

Supplemental Information

Supplemental Information 1 Functional and pathway analysis data from DAVID database.

Click here for additional data file.

Supplemental Information 2 Microarray data from GSE107789.

Click here for additional data file.

Supplemental Information 3 Protein–protein interaction data obtained from STRING database.

Click here for additional data file.

Supplemental Information 4 Network analysis of protein–protein interaction output by Cytoscape.

Click here for additional data file.

Supplemental Information 5 Ct value of qRT-PCR of in vivo and in vitro verification.

Click here for additional data file.

Supplemental Information 6 Raw data of Oil Red O staining.

Sheet 1: Oil Red O staining area and Total area of proximal tibia sections of mice calculated by ImageJ software (n = 4). Sheet 2: Oil Red O staining positive cell counts and total cell counts of BMSCs cultured in growth medium or adipogenic induction medium (n = 3). Five fields were randomly selected and counted for each culture well.

Click here for additional data file.

Additional Information and Declarations

Competing Interests

Author Contributions

Animal Ethics

Data Availability

The authors declare that they have no competing interests.

Pengyu Chen conceived and designed the experiments, performed the experiments, analyzed the data, prepared figures and/or tables, authored or reviewed drafts of the paper, and approved the final draft.

Mingrui Song performed the experiments, prepared figures and/or tables, and approved the final draft.

Yutian Wang analyzed the data, prepared figures and/or tables, and approved the final draft.

Songyun Deng performed the experiments, authored or reviewed drafts of the paper, and approved the final draft.

Weisheng Hong performed the experiments, prepared figures and/or tables, and approved the final draft.

Xianrong Zhang conceived and designed the experiments, authored or reviewed drafts of the paper, supervision, and approved the final draft.

Bin Yu conceived and designed the experiments, authored or reviewed drafts of the paper, supervision, Funding acquisition, and approved the final draft.

The following information was supplied relating to ethical approvals (i.e., approving body and any reference numbers):

Animal Care and Use Committee at the Southern Medical University Nanfang Hospital provided full approval for this research (NFYY-2018-34).

The following information was supplied regarding data availability:

The raw data are available in the Supplemental Files.

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
