# Peer review of "Identification of key genes of human bone marrow stromal cells adipogenesis at an early stage"

_PeerJ, doi:10.7717/peerj.9484_

## Round 0.1 · original submission · Major Revisions

Your manuscript has been reviewed by two experts in the field. As you can see from their comments below, both of them basically appreciate the value of this work but raise several points for its further improvement. Please read their comments carefully and revise the manuscript accordingly.

·

Basic reporting

The content of this paper is thought to be scientific and straightforward. They performed the experiments based on the previous research.

Experimental design

Using the downloaded GEO database, they performed various analysis such as DDEGs, KEGG pathway, and PPI network analysis. Finally, they confirmed the results obtained by the analysis described above in vivo OVX model mouse. The experimental strategy is simple and the obtained results are informative.

Validity of the findings

They used the data that have already uploaded to the gene profiling data base. That is why I feel the novelty of this manuscript is at average level. However, Their analysis sounds interesting for especially the researchers in the field of mesenchymal cell biology.

Additional comments

In this manuscript, Chen et al., investigated the intrinsic molecular pathway and factors involved in adipogenesis of bone marrow stromal cells. Using the downloaded GEO database, they performed various analysis such as DDEGs, KEGG pathway, and PPI network analysis. Finally, they confirmed the results obtained by the analysis described above in vivo OVX model mouse. The experimental strategy is simple and the obtained results are informative. Before it can be considered for publication, I would like to request some revisions or answers against the following points.

1. In the introduction section (lane 72-73), the author described that “further investigations of …..and balance of osteogenic and adipogenic lineage commitment are needed.”
What the authors identified in this paper is the key genes of human BMSCs adipogenesis. Therefore, these sentences in the introduction section might give us the wrong impression that the authors also investigated key factors involved in the balance of osteogenic and adipogenic lineage commitment.
2. Lane 207, “Fig. 4A-C” should be “Fig. 4A and B”.
3. The authors identified many important genes that were significantly changed in this analysis. However, in vivo experiments, the authors checked the expression of only S1PR3, ACSL1, ZBTB16 and GPC3. Why did the authors choose these 4 genes? Selecting ACSL1 seems reasonable based on the data of GSEA enrichment analysis. How about the others?

Reviewer 2 ·

Basic reporting

In this manuscript, the authors used the existed datasets with gene expression during adipogenesis, and extrancted the differentially expressed genes. Those genes were further analyzed witn GO, KEGG, GSEA analysis and PPI network analysis. The diffential expression of some of designated genes were confirmed by cultured cells and OVX.
Those confirmed genes can have the potentiality of function in adipogenesis.

To be acceppted, following concerning to be solved.

1.
In Fig. 2, The red colored expression seems to be all saturated.
The color gradation should be shown in both of red and blue direction.
The value of relative expression should be shown in color guide.

2.
In Fig. 3, P-value for each GO term presented should also be shown similaly to Fig. 4.

3.
In main text line 206, "Fig. 4A-C" should be "Fig. 5A-C".

4.
In Fig. 5E, color gradation guide with the value should be shown.

5.
In Fig. 6, Since figures were too complecate, it is diffecult to understand the detailed network relationship.
Especially in Fig. 6B, nodes and edges should be presented more clearly.

Experimental design

no comment

Validity of the findings

no comment

---

## Round 0.2 · accepted · Accept

Since both of the original reviewers turned out to be too busy to review your revision, I invited the third reviewer. As you can see from his/her comments below, he/she confirmed that the revision has been done appropriately. I also confirm the changes. Thus, I am happy to accept this manuscript for publishing in PeerJ.

Reviewer 3 ·

Basic reporting

No comment.

Experimental design

No comment.

Validity of the findings

No comment.

Additional comments

All points of the requests have been properly answered and corrected.
I agree to accept this manuscript to be published in this journal.